# Unmasking Racial, Ethnic, and Socioeconomic Disparities in United States Chordoma Clinical Trials: Systematic Review

**DOI:** 10.3390/cancers17020225

**Published:** 2025-01-12

**Authors:** Ali Haider Bangash, Jessica Ryvlin, Vikram Chakravarthy, Oluwaseun O. Akinduro, Patricia L. Zadnik Sullivan, Tianyi Niu, Michael A. Galgano, John H. Shin, Ziya L. Gokaslan, Mitchell S. Fourman, Yaroslav Gelfand, Saikiran G. Murthy, Reza Yassari, Rafael De la Garza Ramos

**Affiliations:** 1Spine Research Group, Montefiore Medical Center, Albert Einstein College of Medicine, Bronx, NY 10467, USA; alihaider2022-010@stmu.edu.pk (A.H.B.); ryvlinj@upmc.edu (J.R.); mfourman@montefiore.org (M.S.F.); ygelfand@montefiore.org (Y.G.); samurthy@montefiore.org (S.G.M.); ryassari@montefiore.org (R.Y.); 2Department of Neurosurgery, The Ohio State University Wexner Medical Center, Columbus, OH 43210, USA; vikram.chakravarthy@osumc.edu; 3Department of Neurosurgery, Mayo Clinic, Jacksonville, FL 32224, USA; akinduro.oluwaseun@mayo.edu; 4Department of Neurosurgery, Brown University, Providence, RI 02912, USA; psullivan4@lifespan.org (P.L.Z.S.); tniu@lifespan.org (T.N.); ziya_gokaslan@brown.edu (Z.L.G.); 5Department of Neurosurgery, University of North Carolina, Chapel Hill, NC 27599, USA; mgalgano@email.unc.edu; 6Department of Neurosurgery, Harvard Medical School, Boston, MA 02115, USA; shin.john@mgh.harvard.edu; 7Department of Orthopedic Surgery, Montefiore Medical Center, Albert Einstein College of Medicine, Bronx, NY 10467, USA; 8Department of Neurological Surgery, Montefiore Medical Center, Albert Einstein College of Medicine, Bronx, NY 10467, USA

**Keywords:** chordoma, diversity, race, socioeconomic status, insurance, employment, vulnerability

## Abstract

Chordoma is an aggressive bone cancer that is hard to treat. To find better treatments, we need clinical trials that are inclusive to all patients. Our research looked at chordoma trials in the United States to see whether they include people from different racial, ethnic, and economic backgrounds. We found that minorities were significantly underrepresented in these trials, and information about patients’ socioeconomic status was altogether missing. This lack of diversity could mean that treatments are not being tested on all the groups they need to help. Our findings highlight the need for more inclusive chordoma research. By improving diversity in clinical trials, we can work towards better treatments and outcomes for all patients with chordoma, regardless of their background.

## 1. Introduction

Chordoma is a rare and aggressive type of primary bone tumor that originates from remnants of the notochord, an embryonic structure which guides the formation of the spine. It most commonly arises from the base of the skull (clival chordomas), the vertebral bodies of the mobile spine, or the sacrum [1]. They are slow-growing but locally aggressive tumors, with a tendency to recur and metastasize over time [2]. The clinical presentation varies depending on tumor location, with common symptoms including pain, neurological deficits, and, in sacral chordomas, bowel and bladder dysfunction [3].

Its incidence is estimated around 0.08 cases per 100,000 people [4]. The management of chordoma is challenging, typically requiring a multidisciplinary approach including surgery, radiation therapy, and, increasingly, targeted therapies [5]. Previous studies have suggested potential disparities in the incidence and outcomes of chordoma based on race, ethnicity, and socioeconomic status [6,7]. However, the evidence does not suggest any racial or ethnic predisposition to developing chordoma [8]. There is a lack of clarity regarding the representation of diverse populations in clinical trials investigating chordoma treatments.

Ensuring the adequate representation of diverse patient populations in clinical research is crucial for generating findings that are generalizable and relevant to all those affected by the disease. Nieblas-Bedolla E. et al. examined diversity among pediatric patients with primary central nervous system tumors in the United States and identified that White children may have a higher likelihood of being diagnosed with these tumors, while Hispanic children tend to present with advanced-stage disease and experience poorer outcomes, highlighting notable racial and ethnic disparities [9]. The underrepresentation of certain racial, ethnic, and socioeconomic groups in clinical trials can lead to biased conclusions and hinder the development of equitable treatment strategies [10,11]. This is particularly important for rare diseases like chordoma, where the available patient pool is already limited.

The objective of this study was to evaluate the reporting of racial, ethnic, and socioeconomic diversity in clinical trials exploring treatments for chordoma and compare the reported data with US national data. By assessing the current state of diversity inclusion, we aimed to identify areas for improvement and provide recommendations to enhance the inclusiveness of future chordoma research.

## 2. Materials and Methods

### 2.1. Search Strategy

This systematic review was carried out in accordance with the Preferred Reporting Items for Systematic Review and Meta-Analysis (PRISMA) guidelines [12], with a prospective study protocol (PSP) guiding the objectives, search strategy, and planned analyses developed and subsequently adopted rigorously. Being a systematic review of published trials, this study was exempt from seeking ethics approval.

The literature search was carried out independently by A.H.B. and J.R. through the PubMed/Medline, Cochrane Database of Systematic Reviews (CDSR), Epistemonikos, and ClinicalTrials.gov databases in accordance with the PSP, specifically looking for published chordoma trials undertaken in the United States. The reference lists of the selected eligible studies were also independently searched for relevant publications.

We aimed to explore the reporting of racial, ethnic, and socioeconomic indicators in chordoma trials being undertaken in the United States. Moreover, we also aimed to explore the racial and ethnic diversity of the included patient population in such trials.

The search strategy included the terms “Chordoma” [MeSH terms]. It is concisely summarized in the “Appendix A”.

### 2.2. Study Selection

The cohort of eligible articles was independently reviewed by A.H.B. and V.C. with consideration of the abstracts and full texts, as required. Studies were included if they met the following pre-determined inclusion criteria: (1) clinical trials exploring chordoma with published results; (2) trials undertaken in the USA; and (3) results published from database inception until 19 August 2024. Studies were excluded if they (1) included a patient population other than patients with chordoma, (2) had been undertaken in countries other than the USA, (3) were trial protocols, (4) had no results published, or (5) were review articles, editorials, or conference abstracts.

Any conflict in screening was resolved via consultation with the senior author (R.D.G.R.). The updated PRISMA flow diagram was adopted to represent the study selection process transparently.

Manual extraction of the required data, in accordance with the pre-determined “Characteristics of studies” table, was independently carried out by A.H.B. and O.O.A. The extracted variables included the year of study, the number of patients, and their sex. Racial and ethnic data were also collected when available and defined according to the Food and Drug Administration reporting guidelines. The race categories included White, Black or African American, Asian, American Indian or Alaska Native, Native Hawaiian or Other Pacific Islander, more than one race, and unknown or not reported. Ethnicity categories included Hispanic or Latino, Not Hispanic or Latino, and unknown or not reported. Measures of socioeconomic status were also extracted when available and included insurance status, income, employment status, occupation, primary language, Social Vulnerability Index, Area Deprivation Index, and any other(s).

### 2.3. Quality Assessment and Risk of Bias

The Methodological Index for Non-Randomized Studies (MINORS) tool was independently adopted by A.H.B. and P.L.Z.S. to assess the methodological quality and risk of bias of the included non-randomized single-arm trials [13]. For the MINORS analysis, a global ideal score of 16 for non-comparative studies was recognized. A score of 4 was considered to be very low quality, 5–8 low quality, 9–12 moderate quality, and 13–16 high quality [14]. For the included randomized trial(s), the Revised Cochrane Risk-of-Bias Tool for Randomized Trials (RoB2) was independently adopted by A.H.B. and T.N. to appreciate the methodological quality and risk of bias [15]. The RoB2 analysis allowed for the assessment of the overall risk of bias, stratified in 5 domains: (1) randomization process; (2) deviations from the intended interventions; (3) missing outcome data; (4) measurement of the outcome; and (5) selection of the reported result.

### 2.4. Statistical Analysis

An exploratory data analysis was performed. The categorical variables were expressed as percentages of the total. Pie charts were developed to express the variability reported in different racial and ethnic groups. The collated racial and ethnic diversity reported in the included trials were independently compared by A.H.B., T.N., and J.H.S. with the racial and ethnic diversity of the USA population, as reported in recent data by the US Census Bureau [16]. The N-1 Chi-squared (χ^2^) test was independently implemented by A.H.B. and J.H.S. to determine the statistical significance of the differences in the respective proportions. A double-tailed *p*-value < 0.05 was recognized as statistically significant.

## 3. Results

A search carried out through the PubMed/Medline, CDSR, Epistemonikos, and ClinicalTrials.gov databases yielded 95 hits. After removing 22 duplicates, the abstracts and, where required, the full texts of 74 documents were considered in accordance with the aforementioned inclusion eligibility criteria. Five trials satisfied the inclusion eligibility criteria and were, therefore, included in this review [17,18,19,20,21]. (Table 1) The underlying causes of exclusion are indicated in the relevant component of the PRISMA flow diagram in Figure 1. The complete list of the excluded documents is shared in the “Appendix A”.

### 3.1. General Patient Demographics and Trial Characteristics

The trials were published between 2016 and 2024 and reported on a total of 111 patients with a median age of 63 years (range: 30–83 years), 34% (*n* = 38) of whom were female. A total of 80% (*n* = 4) of the included trials were single-arm non-randomized trials [17,18,19,20,21], whereas 20% (*n* = 1) of the included trials were randomized controlled designs [19]. A total of 60% (*n* = 3) of trials were Phase I [17,20,21], whereas 40% (*n* = 2) were Phase II [18,19]. Systemic and radiation therapies were concurrently explored in 60% (*n* = 3) of the trials [18,19,20], whereas systemic therapy was exclusively explored in 40% (*n* = 2) of the trials [17,21]. Brachyury-based vaccines were the most common systemic therapy, explored in 60% (*n* = 3) of the trials [18,19,21].

Out of a total of 111 patients, 25.5% (*n* = 28) suffered from skull-base chordoma, 24.5% (*n* = 27) from mobile spine chordoma, and 50% (*n* = 56) from sacral/coccyx chordoma.

### 3.2. Quality Assessment and Risk of Bias

Upon undertaking the MINORS analysis on the included single-arm chordoma trials, 25% (one of four) of the single-arm trials were recognized to be of a high methodological quality [20], whereas 75% (three of four) of the single-arm trials were recognized to be of a moderate methodological quality [17,18,19,21] (Table 2). All single-arm trials were found to determine endpoints appropriate to their respective aim(s). However, only 25% (one of four) of the single-arm trials reported a respective loss to follow-up of less than 5% [20] and prospectively calculated the study size [18].

Upon the implementation of RoB2 analysis on the included randomized controlled trial, it was found to have a high overall risk of bias across both: ‘missing outcome data’ domain and ‘measurement of the outcome’ domain [19] (Figure 2).

### 3.3. Race and Ethnicity

Eighty percent (*n* = 4) of the trials reported racial and ethnic data for a total of 90 patients [17,18,19,20] (Table 1) (Figure 3). The most recent USA Census Bureau racial and ethnic data, reporting on a total population of 334 M, was established as the comparison standard [22].

#### 3.3.1. White/Caucasian Patients

Out of a total of 90 patients reported by the included chordoma trials, 91% (*n* = 82) were White/Caucasian, whereas the most recent USA Census Bureau report recognized the US White/Caucasian population to be 75% of the total. This difference in proportion was found to be significant [difference = 15.8% (95% CI, 8.1158 to 20.1193), χ2 = 12.08, and *p*-value= 0.0005].

#### 3.3.2. Black/African American Patients

Out of a total of 90 patients reported by the included chordoma trials, 2% (*n* = 2) were Black/African American, whereas the most recent USA Census Bureau report recognized the US Black/African American population to be 14% of the total. This difference in proportion was found to be significant [difference = 11.5% (95% CI, 5.9885 to 13.0981), χ2 = 10.067, and *p*-value = 0.0015].

#### 3.3.3. Asian Patients

Out of a total of 90 patients reported by the included chordoma trials, 3% (*n* = 3) were Asian, whereas the most recent USA Census Bureau report recognized the US Asian population to be 6% of the total. However, this difference in proportion was found not to be significant [difference = 3.1% (95% CI, −2.9004 to 5.277), χ2 = 1.44, and *p*-value = 0.2295].

#### 3.3.4. American Indian or Alaska Native Patients

Out of a total of 90 patients reported by the included chordoma trials, 1% (*n* = 1) were American Indian or Alaska Native, whereas the most recent USA Census Bureau report recognized the US American Indian or Alaska Native population to be 1.3% of the total. However, this difference in proportion was found not to be significant [difference = 0.2% (95% CI, −4.7104 to 1.1069), χ2 = 0.028, and *p*-value = 0.867].

#### 3.3.5. Native Hawaiian or Other Pacific Islander Patients

Out of a total of 90 patients reported by the included chordoma trials, 0% (*n* = 0) were Native Hawaiian or Other Pacific Islander, whereas the most recent USA Census Bureau report recognized the US Native Hawaiian or Other Pacific Islander population to be 0.3% of the total. However, this difference in proportion was found not to be significant [difference = 0.3% (95% CI, −3.7936 to 0.3019), χ2 = 0.271, and *p*-value = 0.6028].

#### 3.3.6. Unknown or Not Reported Racial Data

Out of a total of 90 patients reported by the included chordoma trials, 2% (*n* = 2) were reported to have an unknown or unreported race, whereas the most recent USA Census Bureau report recognized the proportion of US population of unknown or unreported race to be 0% of the total. This difference in proportion was found to be significant [difference = 2.2% (95% CI, 0.6019 to 7.7115), χ2 = 736,812.78, and *p*-value < 0.0001].

#### 3.3.7. Hispanic or Latino Patients

Out of a total of 90 patients reported by the included chordoma trials, 7% (*n* = 6) were Hispanic or Latino, whereas the most recent USA Census Bureau report recognized the US Hispanic or Latino population to be 20% of the total. This difference in proportion was found to be significant [difference = 12.83% (95% CI, 5.7056 to 16.4069), χ2 = 9.438, and *p*-value = 0.0021].

#### 3.3.8. Non-Hispanic or Latino Patients

Out of a total of 90 patients reported by the included chordoma trials, 92% (*n* = 83) were neither Hispanic nor Latino, whereas the most recent USA Census Bureau report recognized the US non-Hispanic and non-Latino population to be 80.5% of the total. This difference in proportion was found to be significant [difference = 11.7% (95% CI, 4.2783 to 15.6667), χ2 = 7.848, and *p*-value = 0.0051].

#### 3.3.9. Unknown or Not Reported Ethnicity Data

Out of a total of 90 patients reported by the included chordoma trials, 1% (*n* = 1) were either of unknown or unreported ethnicity, whereas the most recent USA Census Bureau report recognized the proportion of US population with unknown or unreported ethnicity to be 0% of the total. This difference in proportion was found to be significant [difference = 1.1% (95% CI, 0.1931 to 6.0104), χ2 = 368,406.379, and *p*-value < 0.0001].

### 3.4. Measures of Socioeconomic Status

Insurance status, income, employment status, occupation, primary language, Social Vulnerability Index, Area Deprivation Index, and other measures of socioeconomic status were not reported by any of the included chordoma trials.

## 4. Discussion

The incidence of chordoma is exceptionally low, with an average age-adjusted incidence in the United States of approximately 0.037 cases per 100,000 male individuals and 0.029 cases per 100,000 female individuals [23,24]. This rarity contributes to the limited understanding of its pathogenesis and optimal treatment approaches. Surgical resection remains a cornerstone of its management, but achieving complete tumor removal while preserving neurological function can be challenging due to the proximity of the tumor to critical structures. Radiation therapy has been employed as an adjuvant to surgery, and newer targeted therapies are emerging as well [25]. Due to their slow growth rate and insidious nature, chordomas often present diagnostic challenges and require specialized management [5]. They have been reported to exhibit variability influenced by several factors, including genetic mutations, tumor location, and microenvironment interactions [26]. However, it is important to note that there is currently no evidence suggesting an inherent genetic predisposition based on different racial or ethnic groups that could explain the observed disparities in representation and outcomes [8]. With race, ethnicity, and socioeconomic factors having been recognized to influence oncologic outcomes, research must encompass diverse patient groups, as only by considering these factors holistically would the development of effective treatments which benefit all patients with chordoma be possible [27].

Our current study revealed that none of the chordoma trials undertaken in the United States provided information on the socioeconomic characteristics of their study populations, limiting our understanding of the generalizability of the reported outcomes. Although race and ethnicity were reported by the included trials, minorities were found to be severely underrepresented. White patients and Non-Hispanic or Latino patients comprised more than 90% of the cohort and were found to be significantly over-represented when compared with US national data. Unfortunately, the lack of diversity in these trials is commensurate with other investigations. Taha et al. conducted a review of brain tumor clinical trials, finding that only 28% of trials with results had published data on race or ethnicity [28]. Furthermore, the authors found that White patients were significantly overrepresented in trials for both high-grade tumors and metastatic brain lesions. Not surprisingly, Black or African American patients, Hispanic or Latino patients, and Asian patients were significantly underrepresented [28].

The underrepresentation of certain racial, ethnic, and socioeconomic groups in chordoma research is concerning, as it may obscure potential disparities in disease incidence, treatment access, and clinical outcomes. Previous studies have suggested that factors such as race, ethnicity, and socioeconomic status can influence the epidemiology and management of chordoma. Elsamadicy et al. found that surgical intervention of primary osseous tumors of the spine was highest for White and Hispanic patients compared to Black or African American patients [29]. Additionally, the five-year survival was reported to be the highest in White patients compared to other patient populations [29]. Lee et al. examined the California Cancer Registry and reported that, after adjustment for clinically relevant factors, a high socioeconomic status was significantly associated with a longer survival in patients with chordoma [30].

These findings align with a recent study by Battistin U et al. that analyzed the impact of socioeconomic determinants on access to care and survival in patients with spinal chordoma using National Cancer Database data [31]. Their study, which included 1769 patients, found that Black/African American patients had a significantly lower likelihood of undergoing surgery compared to other racial groups. Additionally, they reported higher survival probabilities among patients with other government insurances, higher income levels, residing in metropolitan areas, and receiving care at academic/research centers. Conversely, a lower survival was observed in uninsured patients, those living in rural areas, and those treated at community cancer programs [31]. Failing to capture these important patient-level factors in clinical trials could hinder the development of tailored treatment strategies and perpetuate existing healthcare inequities.

Multiple factors could likely have contributed to the observed underrepresentation of minorities in chordoma clinical trials, with the potential involvement of structural, clinical, and sociocultural barriers. Major academic centers conducting trials, especially those for rare diseases, are often concentrated in urban areas, potentially limiting access for rural populations where minorities may be concentrated [32]. The lack of insurance coverage for clinical trial participation and transportation costs, as well as the potential loss of work time could disproportionately affect minority populations [33]. The limited availability of trial materials in multiple languages and insufficient interpreter services may also exclude non-English speaking participants [34].

Moreover, restrictive inclusion criteria may disproportionately exclude minority populations with higher rates of comorbidities [17,18]. An implicit bias in healthcare provider referrals and a lack of diversity among treating physicians may also influence trial participation [35]. Furthermore, the healthcare providers serving minority communities could have less exposure to or information about available clinical trials, leading to the underrepresentation of minority racial–ethnic groups [36]. In addition to this, past unethical research practices could have created lasting skepticism about clinical research participation among certain minority communities, and different cultural perspectives about medical research and traditional medicine may influence willingness to participate [37,38]. Furthermore, disparities in health education and understanding of clinical trials may affect informed decision making about participation [39].

Adding to this, the lack of adequate health insurance may also affect both initial access to specialized care and subsequent trial participation [40]. These barriers are often interconnected and cumulative in their impact. For example, a limited English proficiency may compound difficulties in understanding trial information, while financial constraints may make multiple visits to distant trial sites impractical. Additionally, the rare nature of chordoma adds another layer of complexity, as specialized treatment centers may be geographically concentrated, potentially exacerbating access disparities for minority populations.

Attempts have been undertaken to further identify barriers to inclusivity in clinical trials. A qualitative investigation involving stakeholders from Switzerland, Germany, and Canada examined the causes of recruitment failures in clinical trials [41]. The study identified overly optimistic recruitment projections, excessively restrictive eligibility criteria, lack of recruiter and trial team engagement, recruiters’ lack of skills, training, and experience, inadequate initial funding, and excessive participant burden as primary obstacles [41]. A thorough study elaborated on the Diverse and Equitable Participation in Clinical Trials (DEPICT) Act, which was enacted to address challenges and promote inclusivity within clinical trials [42]. Diversity in clinical trials was suggested to be increased by providing pre-planned translated materials for patients with a non-English primary language and offering financial support through adequate participant compensation and travel reimbursement. Additionally, targeted marketing, community outreach, and the enhanced diversity of study staff were proposed to help reach underrepresented populations and improve inclusivity in research [42].

Therefore, to address these roadblocks, researchers conducting chordoma trials should make a concerted effort to recruit diverse patient populations and thoroughly report on the sociodemographic characteristics of their study cohorts. This may involve targeted outreach to underserved communities, collaborations with healthcare providers serving diverse patient populations, and the implementation of inclusive recruitment strategies [43]. Additionally, journal editors and funding agencies should consider mandating the reporting of diversity metrics as a prerequisite for publication or grant approval [44]. Several methodological approaches can enhance trial generalizability while maintaining scientific rigor. These include adopting pragmatic trial elements, implementing broader eligibility criteria that do not unnecessarily exclude minority populations, and utilizing adaptive trial designs that allow for more flexible participant allocation [45,46]. Statistical approaches such as pre-planned subgroup analyses by race, ethnicity, and socioeconomic factors, along with Bayesian methods to leverage existing data, can also provide more nuanced insights across different populations [47,48]. The integration of real-world evidence through hybrid trial designs and comprehensive patient registries can further enhance result generalizability [49]. Implementation strategies should also include cultural competency training for research staff, multilingual study materials, flexible scheduling options, and practical support such as transportation assistance [50,51]. While these approaches may increase operational complexity, the resulting improvements in generalizability and clinical relevance justify such investments, particularly in rare diseases such as chordoma. In order to fund these diversity and inclusivity efforts, funding agencies should create dedicated mechanisms for diversity-focused research with incentive programs reserved for trials meeting pre-specified diversity targets, as well as priority support for infrastructure development in underserved communities [52,53,54].

Although this study had several strengths, including an exhaustive literature search across databases and a robust statistical analysis via comparison with the most recent standardized data, its limitations should also be acknowledged. The most important limitation of this study was the low number of trials meeting the inclusion criteria, representing a very small number of patients. This may have reduced the statistical power of our analyses, potentially affecting our ability to detect smaller but meaningful differences in racial and ethnic representation. This shortcoming must be addressed in future studies. However, it is important to contextualize this limitation within the reality of chordoma’s extreme rarity. Furthermore, although the included trials were conducted in major metropolitan areas, the causes for a lack of diversity in trial enrollment might not have been fully elucidated. The potential for publication bias should also be recognized, as trials with an incomplete or absent reporting of diversity data may be less likely to be published. Furthermore, the retrospective nature of our analysis precluded the assessment of the reasons underlying the observed gaps in diversity reporting. Despite these limitations, our systematic review provides valuable initial insights into the current state of diversity in USA chordoma trials and highlights areas requiring attention in future research.

## 5. Conclusions

This systematic review revealed the significant underrepresentation of minorities in chordoma clinical trials. Furthermore, there was no reporting of socioeconomic data. Addressing these issues is crucial to ensuring the generalizability of research findings and promoting equitable access to novel chordoma treatments. Concerted efforts by researchers, clinicians, and policymakers are needed to enhance the inclusivity of future chordoma clinical trials and ultimately improve outcomes for all those affected by this rare and challenging disease.

## Figures and Tables

**Figure 1 cancers-17-00225-f001:**
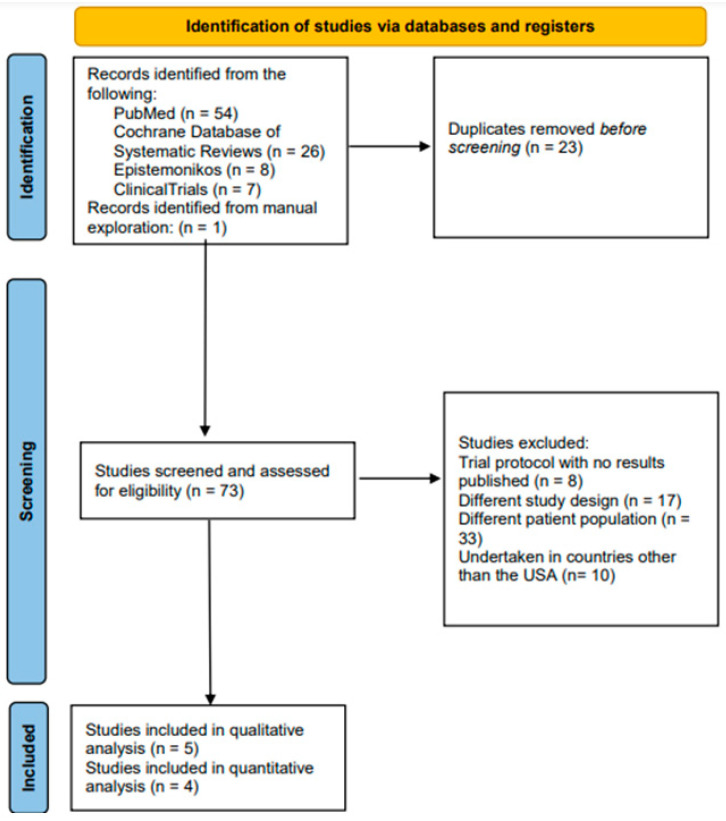
PRISMA flow-chart illustrating the search, screening, and inclusion of published US clinical trials exploring the management of chordoma.

**Figure 2 cancers-17-00225-f002:**
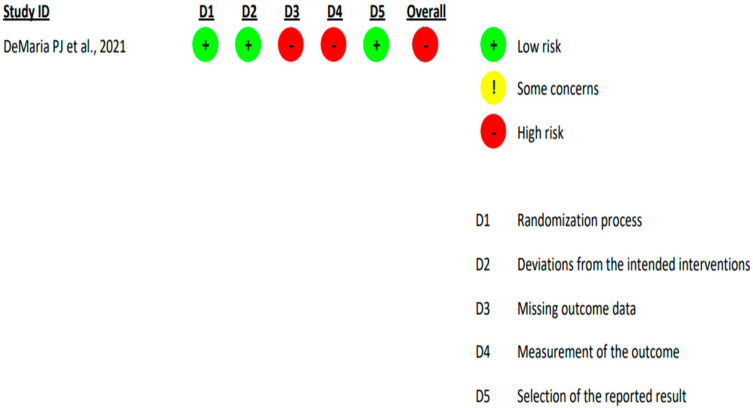
Revised Cochrane Risk-of-Bias Tool for Randomized Trials (RoB2) analysis of published US randomised-controlled clinical trial [DeMaria PJ et al., 2021] [19] exploring the management of chordoma.

**Figure 3 cancers-17-00225-f003:**
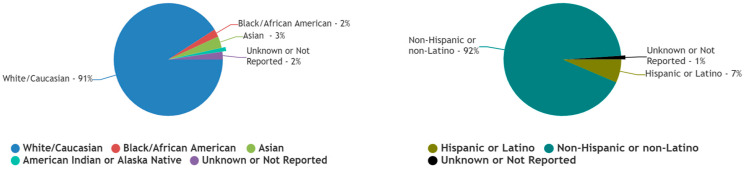
Comparative racial and ethnic inclusion of patients reported in published US clinical trials exploring the management of chordoma.

**Table 1 cancers-17-00225-t001:** The table details the particulars of published US clinical trials exploring the management of chordoma.

Study and Year	Patients	Age	Race and Ethnicity of Patients	Intervention	Outcome
Kesari S et al., 2024 [17]	15	Median age: 61 years (Range: 30–80)	Race: 86.7% White (*n* = 13) and 13.3% Asian (*n* = 2) Ethnicity: 86.7% Not Hispanic or Latino (*n* = 13) and 3.2% Hispanic or Latino (*n* = 2)	Pemetrexed	Median Progression-free survival (PFS) = 10.5 months 6-month PFS = 67% Stable disease = 10/14 participants (71%)
Bavarian Nordic, 2023 [18]	29	Mean age: 65.9 (10.82)	Race: 89.7% White (*n* = 26), 3.4% Black or African American (*n* = 1), and 6.9% Unknown or Not Reported (*n* = 2) Ethnicity: 89.7% Not Hispanic or Latino (*n* = 26), 6.9% Hispanic or Latino (*n* = 2), and 3.4% Unknown or Not Reported (*n* = 1)	BN Brachyury + Standard-of-Care Radiotherapy	Objective Response Rate (ORR) = 7.7 (95% CI, 2.6 to 20.8)
DeMaria PJ et al., 2021 [19]	24	Median age: 61 years (Range: 30–76)	Race: 87.5% White (*n* = 21), 4.2% Black or African American (*n* = 1), 4.2% Asian (*n* = 1), and 4.2% American Indian or Alaska Native (*n* = 1) Ethnicity: 95.8% Not Hispanic or Latino (*n* = 23) and 4.2% Hispanic or Latino (*n* = 1)	Yeast–Brachyury Vaccine (GI-6301) + Standard-of-Care Radiotherapy	Overall Response Rate = vaccine arm 9% (1/11 patients) and placebo arm 8% (1/13 patients) PFS = vaccine arm median 20.6 months (95% CI, 5.7 to 37.5 months) and placebo arm median 25.9 months (95% CI, 9.2 to 30.8 months) OS = vaccine arm median 37.5 months (95% CI, 21.6 to 50.6 months) and placebo arm median not reached
Cote GM et al., 2018 [20]	22	Median age: 65 years (Range: 30–83)	Race: 100% White (*n* = 22) Ethnicity: 95.5% Not Hispanic or Latino (*n* = 21) and 4.5% Hispanic or Latino (*n* = 1)	Nilotinib + Standard-of-Care Radiotherapy	ORR: 6% (1/18 patients, 95% CI, and 0.1% to 27%) PFS: median 58.15 months (95% CI, 39.10 to N) OS: median 61.5 months (95% CI, 43.1 to N) and 2-year OS rate of 95%
Fenerty KE et al., 2016 [21]	21	Median age: 60 years (Range: 32–82)	Race: Not reported Ethnicity: Not reported	Yeast–Brachyury Vaccine (GI-6301), MVA–Brachyury–TRICOM Vaccine	Time to Progression (TTP) = good clinical outcome group with a longer TTP by volumetric assessment (*p* = 0.012, HR 0.21, and *p* = 0.02) and a poor clinical outcome group with a shorter TTP by volumetric assessment No significant difference in the TTP between groups when assessed by RECIST criteria (*p* = 0.37, HR 0.52, and *p* = 0.38)

PFS = progression-free survival; OS = overall survival; HR = hazard ratio; ORR = objective response rate; RECIST = Response Evaluation Criteria in Solid Tumors; TTP = time to progression; VEGF = vascular endothelial growth factor; IQR = inter-quartile range; FMISO = fluoromisonidazole; and T/C = tumor/cerebellum ratio.

**Table 2 cancers-17-00225-t002:** Methodological Index for Non-Randomized Studies (MINORS) analysis of published US clinical trials exploring the management of chordoma.

Evaluation Parameters	Trials			
	Kesari S et al., 2024 [17]	Bavarian Nordic, 2023 [18]	Cote GM et al., 2018 [20]	Fenerty KE et al., 2016 [21]
A clearly stated aim	2	2	2	2
Inclusion of consecutive patients	1	1	2	1
Prospective collection of data	2	2	2	1
Endpoints appropriate to the aim of the study	2	2	2	2
Unbiased assessment of the study endpoint	1	1	2	1
Follow-up period appropriate to the aim of the study	2	1	2	2
Loss to follow-up less than 5%	1	0	2	1
Prospective calculation of the study size	1	2	1	0
Scores	12	10	15	10

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
