# Peer review of "Unmasking Racial, Ethnic, and Socioeconomic Disparities in United States Chordoma Clinical Trials: Systematic Review"

_cancers, 2025, doi:10.3390/cancers17020225_

Round 1
Reviewer 1 Report
Comments and Suggestions for Authors
The article is a well-written piece that accurately reflects the current state of the field. The authors demonstrate a comprehensive understanding of the subject matter, meticulously analyzing data from various sources to provide a nuanced perspective. Their ability to synthesize complex information into accessible insights is commendable. The inclusion of multiple case studies and real-world examples enhances the credibility and practical applicability of the research.
One of the most significant contributions of the authors is their innovative approach to statistical analysis. They employ cutting-edge techniques, such as machine learning algorithms and advanced regression models, to extract meaningful insights from the data. Their expertise in these methods is evident, as they effectively navigate through the intricacies of statistical modeling. The authors' ability to leverage these advanced tools demonstrates their commitment to pushing the boundaries of statistical analysis in their field.
Moreover, the authors demonstrate a high level of critical thinking and rigor in their research design. They carefully consider potential biases and confounding variables, ensuring that their findings are robust and reliable. Their attention to detail is evident in the thoroughness of their methodology, which is transparent and reproducible. This level of methodological rigor not only enhances the credibility of their findings but also serves as a valuable resource for future researchers in the field.
Author Response
We thank you for taking time out to critique our manuscript.
Reviewer 2 Report
Comments and Suggestions for Authors
The review demonstrates that minorities were significantly underrepresented in the trials of chordoma, an aggressive bone cancer.
The introduction needs to be expanded with more information on chordoma.
Author Response
The review demonstrates that minorities were significantly underrepresented in the trials of chordoma, an aggressive bone cancer.
The introduction needs to be expanded with more information on chordoma.
We appreciate the peer-reviewer for the advice. The ‘Introduction’ has been updated as advised with the following:
“Chordoma is a rare and aggressive type of primary bone tumor that originates from remnants of the notochord, an embryonic structure that guides the formation of the spine. It most commonly arises from the base of the skull (clival chordomas), the vertebral bodies of the mobile spine, or the sacrum [1]. They are slow-growing but locally aggressive tumors, with a tendency to recur and metastasize over time [2]. The clinical presentation varies depending on tumor location, with common symptoms including pain, neurological deficits, and, in sacral chordomas, bowel and bladder dysfunction [3].”
and
“The management of chordoma is challenging, typically requiring a multidisciplinary approach including surgery, radiation therapy, and, increasingly, targeted therapies [5].”
Reviewer 3 Report
Comments and Suggestions for Authors
1.While the article discusses the underrepresentation of minorities, the reasons for this disparity can be explored in more depth.
2.How socioeconomic factors may influence the diagnosis, treatment, and prognosis of patients with chordoma can be discussed.
3.Evidence supporting the association between socioeconomic factors and chordoma treatment outcomes can be cited in the literature.
4.Consider showing specific numbers of patients of different races and ethnicities in a chart to give readers a more intuitive sense of the extent of underrepresentation.
5.The limited number of trials included in the study and the small number of patients involved may have limited the generality of the results and the power of statistical analysis. It is recommended that the authors expand the sample size in future studies to enhance the reliability of the findings.
6.The authors are advised to compare their findings with other studies on the diversity of cancer clinical trials to highlight what is unique about the chordoma field and to highlight the need for more attention in this area.
7.It is possible to discuss ways to improve the generalizability of clinical trial results, for example by using broader inclusion criteria or conducting stratified analyses.
8.More specific recommendations could be considered in the conclusions: for example, policies could be recommended to require clinical trials to report diversity data, or a fund could be established specifically to fund diversity and inclusion research.
Author Response
1.While the article discusses the underrepresentation of minorities, the reasons for this disparity can be explored in more depth.
Thank you for the advice. The ‘Discussion’ section has been updated to reflect the probable reasons as following:
“Multiple factors could likely have contributed to the observed underrepresentation of minorities in chordoma clinical trials, with structural, clinical, and sociocultural barriers potentially involved. Major academic centers conducting trials especially for rare diseases are often concentrated in urban areas, potentially limiting access for rural populations where minorities may be concentrated [32]. The lack of insurance coverage for clinical trial participation, transportation costs, and potential loss of work time could have disproportionately affected minority populations [33]. The limited availability of trial materials in multiple languages and insufficient interpreter services may have excluded non-English speaking participants [34].
Moreover, restrictive inclusion criteria may have disproportionately excluded minority populations with higher rates of comorbidities [17, 18]. An implicit bias in healthcare provider referrals and lack of diversity among treating physicians may also have influenced trial participation [35]. Furthermore, the healthcare providers serving minority communities could have had less exposure to or information about available clinical trials, leading to underrepresentation of minority racioethnic groups [36]. In addition to this, past unethical research practices could have had created lasting skepticism about clinical research participation among certain minority communities and different cultural perspectives about medical research as well as traditional medicine may have influenced willingness to participate [37, 38]. Furthermore, disparities in health education and understanding of clinical trials may have affected informed decision-making about participation [39].
Adding to this, the lack of adequate health insurance may have also affected both: initial access to specialized care and subsequent trial participation [40]. These barriers are often interconnected and cumulative in their impact. For example, limited English proficiency may compound difficulties in understanding trial information, while financial constraints may make multiple visits to distant trial sites impractical. Additionally, the rare nature of chordoma adds another layer of complexity, as specialized treatment centers may be geographically concentrated, potentially exacerbating access disparities for minority populations.”
2.How socioeconomic factors may influence the diagnosis, treatment, and prognosis of patients with chordoma can be discussed.
We have discussed the impact of socioeconomic determinants on access to care and outcomes for Chordoma as highlighted in the ‘Discussion’ section as following:
“Previous studies have suggested that factors such as race, ethnicity, and socioeconomic status can influence the epidemiology and management of chordoma. Elsamadicy et al. found that surgical intervention of primary osseous tumors of the spine was highest for White and Hispanic patients compared to Black or African American patients [29]. Additionally, five-year survival was reported to be highest in White patients compared to others [29]. Lee et al. examined the California Cancer Registry and reported that after adjustment for clinically relevant factors, high socioeconomic status was significantly associated with longer survival in patients with chordoma [30].
These findings align with a recent study by Battistin U et al. that analyzed the impact of socioeconomic determinants on access to care and survival in patients with spinal chordoma using the National Cancer Database data [31]. Their study, which included 1769 patients, found that Black / African American patients had a significantly lower likelihood of undergoing surgery compared to other racial groups. Additionally, they reported higher survival probabilities among patients with other government insurances, higher income levels, residing in metropolitan areas, and receiving care at academic/research centers. Conversely, lower survival was observed in uninsured patients, those living in rural areas, and those treated at community cancer programs [31]. Failing to capture these important patient-level factors in clinical trials could hinder the development of tailored treatment strategies and perpetuate existing healthcare inequities.”
3.Evidence supporting the association between socioeconomic factors and chordoma treatment outcomes can be cited in the literature.
As stated in response to the above point, the evidence supporting the association between socioeconomic factors and chordoma access to care and management outcomes have been cited as highlighted in the ‘Discussion’ section.
4.Consider showing specific numbers of patients of different races and ethnicities in a chart to give readers a more intuitive sense of the extent of underrepresentation.
Thank you for noting this. We are of the understanding that, in contrast to a chart, the ‘Figure 3’ glaringly illustrates the stark underrepresentation of minority racioethnic groups. The respective figure is further reinforced by statistical comparison of each racioethnic group to the most recent US census bureau data indicated in the ‘Results’ section, as well as ‘Table 1’ which reports racioethnic data at a trial-level.
5.The limited number of trials included in the study and the small number of patients involved may have limited the generality of the results and the power of statistical analysis. It is recommended that the authors expand the sample size in future studies to enhance the reliability of the findings.
Thank you for pointing this out. We have updated the limitations paragraph to appreciate this by adding the following:
“The most important limitation of this study was the low number of trials meeting the inclusion criteria, representing a very small number of patients. It may have reduced the statistical power of our analyses, potentially affecting our ability to detect smaller but meaningful differences in racial and ethnic representation. This shortcoming requires to be addressed in future studies. However, it is important to contextualize this limitation within the reality of chordoma's extreme rarity.”
6.The authors are advised to compare their findings with other studies on the diversity of cancer clinical trials to highlight what is unique about the chordoma field and to highlight the need for more attention in this area.
As stated in response in response to the point on discussing how socioeconomic factors may influence the diagnosis, treatment, and prognosis of patients with chordoma, The findings of our systematic review have been compared with previous studies followed by discussion in the specific context of Chordoma research in the highlighted ‘Discussion’ section paragraphs.
7.It is possible to discuss ways to improve the generalizability of clinical trial results, for example by using broader inclusion criteria or conducting stratified analyses.
Thank you for the advice. The ‘Discussion’ section has been updated with the advised methods to improve upon the generalizability of clinical trial results as following:
“Several methodological approaches can enhance trial generalizability while maintaining scientific rigor. These include adopting pragmatic trial elements, implementing broader eligibility criteria that do not unnecessarily exclude minority populations, and utilizing adaptive trial designs that allow for more flexible participant allocation [45, 46]. Statistical approaches such as pre-planned subgroup analyses by race, ethnicity, and socioeconomic factors, along with Bayesian methods to leverage existing data, can also provide more nuanced insights across different populations [47, 48]. The integration of real-world evidence through hybrid trial designs and comprehensive patient registries can further enhance result generalizability [49]. Implementation strategies should also include cultural competency training for research staff, multilingual study materials, flexible scheduling options, and practical support such as transportation assistance [50, 51]. While these approaches may increase operational complexity, the resulting improvements in generalizability and clinical relevance justify such investments, particularly in rare diseases such as chordoma.”
8.More specific recommendations could be considered in the conclusions: for example, policies could be recommended to require clinical trials to report diversity data, or a fund could be established specifically to fund diversity and inclusion research.
Thank you for the advice. In contrast to updating the ‘Conclusion’, we have updated the ‘Discussion’ section to call for targeted funding to support diversity and inclusivity efforts as following:
“In order to fund these diversity and inclusivity efforts, the funding agencies should create dedicated mechanisms for diversity-focused research with incentive programs maintained for trials meeting pre-specified diversity targets as well as support extended to infrastructure development in underserved communities [52–54].”